# *Sargassum fusiforme* Polysaccharide-Based Hydrogel Microspheres Enhance Crystal Violet Dye Adsorption Properties

**DOI:** 10.3390/molecules27154686

**Published:** 2022-07-22

**Authors:** Bingxue Lv, Jiahao Ren, Yang Chen, Siyu Guo, Minqian Wu, Lijun You

**Affiliations:** School of Food Science and Engineering, South China University of Technology, Guangzhou 510640, China; 201992393124@mail.scut.edu.cn (B.L.); 201992393125@mail.scut.edu.cn (J.R.); 202030311488@mail.scut.edu.cn (Y.C.); 201930210280@mail.scut.edu.cn (S.G.); 202064311188@mail.scut.edu.cn (M.W.)

**Keywords:** *Sargassum fusiforme*, polysaccharides, hydrogel microsphere, sodium alginate, crystal violet dye, absorption

## Abstract

In this study, a polysaccharide-based hydrogel microsphere (SFP/SA) was prepared using *S. fusiforme* polysaccharide (SFP) and sodium alginate (SA). Fourier transform infrared spectroscopy (FT-IR) demonstrated that SFP was effectively loaded onto the hydrogel microsphere. Texture profile analysis (TPA) and differential scanning calorimetry (DSC) showed that, with the increase of SFP concentration, the hardness of SFP/SA decreased, while the springiness and cohesiveness of SFP/SA increased, and the thermal stability of SFP/SA improved. The equilibrium adsorption capacity of SFP/SA increased from 8.20 mg/g (without SFP) to 67.95 mg/g (SFP accounted 80%) without swelling, and from 35.05 mg/g (without SFP) to 81.98 mg/g (SFP accounted 80%) after 24 h swelling. The adsorption of crystal violet (CV) dye by SFP/SA followed pseudo-first order and pseudo-second order kinetics (both with R^2^ > 0.99). The diffusion of intraparticle in CV dye was not the only influencing factor. Moreover, the adsorption of CV dye for SFP/SA (SFP accounted 60%) fit the Langmuir and Temkin isotherm models. SFP/SA exhibited good regenerative adsorption capacity. Its adsorption rate remained at > 97% at the 10th consecutive cycle while SFP accounted for 80%. The results showed that the addition of *Sargassum fusiforme* polysaccharide could increase the springiness, cohesiveness and thermal stability of the hydrogel microsphere, as well as improve the adsorption capacity of crystal violet dye.

## 1. Introduction

With the development of global industrialization, different forms of environmental pollution have become a major problem and burden for most public healthcare systems across the world. Among them, water pollution caused by various industries has seriously affected people’s health and life. Crystal violet (CV) has long been widely used in the textile industry and in other light industries. In recent years, it has attracted much attention due to its mutagenicity and mitotic toxicity [1]. Studies have shown that CV can cause symptoms such as increased heartbeat, vomiting, shock, limb twitching and tissue necrosis [2]. When the CV level is high enough, it can hinder the photosynthesis of aquatic organisms and the dissolution of oxygen, destroying the ecosystem [3]. However, CV is a kind of refractory dye molecule and is difficult to be removed since it has a complex chemical structure, able to resist conventional physical, chemical and biological degradation [4]. Therefore, it is necessary to remove CV from the environment.

At present, the commonly used methods for removing dyes from wastewater include electrochemical treatment, precipitation, ion exchange, reverse osmosis and evaporation [5]. However, these traditional methods are disadvantaged by having low adsorption rates and being difficult to recycle. Physical adsorption has attracted a lot of attention due to its wide application, fast response and environmentally friendliness. In recent years, scientists have used a variety of adsorbents to treat dye wastewater such as activated carbon, zeolite, clay, biomass, functional polymer resin and polymer hydrogel [6]. Among these, biodegradable polymer hydrogels are favored by researchers due to their good regenerative adsorption capacities and low cost [7].

*Sargassum fusiforme* is a warm temperate to subtropical seaweed that belongs to the Sargassaceae family of Fucales, Phaeophyta [8]. *Sargassum fusiforme* is widely distributed in China, Japan, Korea and other countries. Polysaccharides extracted from *S. fusiforme* have good biological adaptability and biodegradability, which are widely used in the food and medicine fields [9]. *S. fusiforme* polysaccharide could improve its non-electrostatic interaction when embedded in polymer materials due to its large amount of sulfate groups, and thereby might enhance polymer materials’ absorption dye capacity. Hydrogel microspheres could trap dye molecules and absorb dye solutions due to their electrostatic and hydrophilic interactions during the swelling process.

In this study, *S. fusiforme* polysaccharide (SFP) was added to sodium alginate (SA) for the first time, and prepared as a polysaccharide-based hydrogel microsphere (named as SFP/SA) after crosslinking with CaCl_2_. Then, SFP/SA was used to remove CV by way of their electrostatic interaction. Afterwards, the internal structure and textural properties of SFP/SA prepared with different SFP contents were studied using Fourier transform infrared spectroscopy (FT-IR), scanning electron microscopy (SEM), differential scanning calorimetry (DSC) and texture profile analysis (TPA). Meanwhile, the kinetic and isothermal models of the adsorption of CV by SFP/SA were investigated. The regeneration and reuse performance of SFP/SA for CV adsorption was studied as well.

## 2. Materials and methods

### 2.1. Materials and Chemicals

*S. fusiforme* was collected (August 2020) from Wenzhou, Zhejiang Province, China. *S. fusiforme* was ground by a grinder (FW135, Tianjin Taisite Instrument Co., Ltd., Tianjin, China) and then the ultra-micro powder was obtained using an ultra-micro pulverizer (XDW6-BI, Jinan Tatsu Micro Machinery Co., Ltd., Jinan, China).

CaCl_2_ (analytical reagent grade, Xilong Science Co., Ltd., Shantou, China), boric acid (analytical reagent grade, Maclin Biochemical Reagent Co., Ltd., Shanghai, China), sodium alginate (SA) (analytical reagent grade, McLin Biochemical Reagent Co., Ltd., Shanghai, China), cationic dye crystal violet (CV) (analytical reagent grade, Fuchen Chemical reagent Co., Ltd., Tianjin, China) and ethanol (95%) (analytical reagent grade, Fuyu Fine Chemicals Co., Ltd., Tianjin, China) were used. The water used in the experiment was double distilled water. Spectrophotometric measurement was conducted by using a full wavelength microplate reader (Molecular Devices SpectraMax 190, Aiyan Biotechnology Co., Ltd., Shanghai, China). All chemicals were commercially available and used without further purification.

### 2.2. Preparation of SFP

SFP was prepared from *S. fusiforme* according to our previously reported procedure, with some modifications [10]. The ultra-micro powder of *S. fusiforme* was refluxed three times (3 h, 1 h and 1 h, respectively) with 95% ethanol at a ratio of 1:4 (*w*/*v*, g/mL) to remove lipids and low molecular weight impurities such as pigments. After centrifugation (6800 rpm, 20 min) the residue was collected and dried overnight in an oven at 60 °C. The dried powder was extracted with double distilled water at a ratio of 1:50 (*w*/*v*, g/mL) for 4 h. After suction filtration, the supernatant was concentrated to 1/15 of the original volume using a reducing rotary evaporator (Hei-VAP Value, Heidoph, Nuremberg, Germany) at 60 °C. It was then precipitated with 4 times the volume of 95% ethanol (*v*/*v*) for 12 h at 4 °C. After centrifugation (6800 rpm, 20 min), the precipitate was collected and re-dissolved in double distilled water, and SFP was finally obtained through vacuum freeze-drying (alpha 1-2 LD plus, Martin Christ gefriedrocknungsanlage GmbH, Osterode am Harz, Germany) [11].

### 2.3. Preparation of SFP/SA

The total weight of SFP and SA was 1 g, and the mixture was dissolved in 40 mL of water. It was placed in a constant temperature magnetic stirring pot (ZNCL-DLS 130 × 60, Henan Ebbott Technology Development Co., Ltd., Zhengzhou, Henan, China), and kept at 30 °C at a speed of 500 rpm for 2 h to form a uniform mixed solution. Then, the solution was dropped into 2% calcium chloride-saturated boric acid solution by syringe to obtain a hydrogel microsphere. After soaking in the above solution for 24 h, the hydrogel microsphere was washed 3 times with water. Then SFP/SA with a particle size of 2–4 mm was obtained after freeze-drying for 24 h (Alpha 1-2 LD plus, Christ Company, Osterode, Lower Saxony, Germany). At the same time, samples without SFP were prepared according to the above method. The chemical compositions of the samples are shown in Table 1. Different contents of polysaccharides (0, 20, 40, 60 and 80% SFP) were added into SFP/SA and named as SFP/SA-0, SFP/SA-20, SFP/SA-40, SFP/SA-60 and SFP/SA-80, respectively.

### 2.4. Characterization of SFP/SA

#### 2.4.1. Scanning Electron Microscope Analysis

All SFP/SA samples were placed in water or CV dye at 25 °C and soaked for 24 h. The samples were then immediately frozen with liquid nitrogen to maintain a complete network structure. All SFP/SA after the above treatment were freeze-dried in a vacuum for 24 h, and the cross-sectional morphology of the SFP/SA was characterized by a SEM (EVO 18, Carl Zeiss AG Co., Ltd., Oberkohen, Baden-Wurttemberg, Germany). The samples were observed at an acceleration voltage of 10 kV under vacuum. Micrographs were taken at a magnification of 500× and 2000×.

#### 2.4.2. Fourier Transform Infrared Spectroscopy Analysis

The samples were ground into powder and treated by the KBr tableting method. A FT-IR spectrophotometer (Tensor 27, Bruker Co., Ltd., Bergisch Gladbach, Germany) was used to record samples at the scanning wavelength range of 4000–500 cm^−1^ and the scanning number was 32 times.

#### 2.4.3. Texture Profile Analysis

All SFP/SA were placed in water at 25 °C and soaked for 24 h. TPA was performed using a texture analyzer (TA. XT PlusC, Stable Micro Systems Co., Ltd., Godalming, Surrey, UK). TPA of all SFP/SA adopted a 50 g contact force, which was compressed twice. The compression amount was 75% of the hydrogel microsphere volume. The compression probe was a cylindrical stainless steel probe (P36-R). The velocities before, during and after compression were 5 mm/s, 1 mm/s and 20 mm/s, respectively. All tests were carried out at 25 °C and repeated six times. The hardness, springiness, cohesiveness and resilience of SFP/SA samples were measured. Parameters were calculated by Texture Expert software.

#### 2.4.4. Differential Scanning Calorimetric Analysis

A total of 10 mg dried SFP/SA was accurately weighed and placed in a sample crucible. The SFP/SA were tested by DSC (DSC250, TA Instruments Co., Ltd., Newcastle, PA, USA) at a heating rate of 20 °C/min under N_2_ flow, and a measurement range of 25–400 °C.

### 2.5. Absorption and Separation of CV by SFP/SA

#### 2.5.1. Absorption of CV

Different SFP/SA samples with the same weight, 25 mg, were immersed in 25 mL of CV solution (original concentration 100 mg/L). The SFP/SA were filtered and separated after adsorption saturation. The residual concentration of CV in the solution was determined by a full-wavelength microplate reader at 585 nm.

The equilibrium adsorption capacity Q_e_ (milligrams of dyes per gram of dry gel) is calculated by the following equation [12]:(1)$$Q_{e}=\frac{C_{0}{\;-\;C}_{e}V}{m}$$
where C_0_ (mg/L) and C_e_ (mg/L) are the initial and final concentrations of CV solution, respectively. V (L) and m (mg) are the solution volume and the SFP/SA mass, respectively.

The absorption capacity Q_t_ (milligrams of dyes per gram of dry gel) is calculated by the following equation:(2)$$Q_{t}=\frac{C_{0}\;{-\;C}_{t}V}{m}$$
where C_0_ (mg/L) and C_t_ (mg/L) are the initial and t time concentrations of CV solution, respectively. V (L) and m (mg) are the solution volume and the SFP/SA mass, respectively.

The removal percentage is obtained using the following equation [13]:(3)$$\mathrm{removal}\;\mathrm{percentage}\;(\%)=\frac{C_{0}\;{-\;C}_{e}}{C_{0}}\times 100\%$$
where C_0_ (mg/L) and C_e_ (mg/L) are the initial and equilibrium concentrations of CV solution, respectively.

#### 2.5.2. Adsorption Kinetic Model of CV by SFP/SA

The adsorption kinetics affect the adsorption rate of CV by SFP/SA and determine the required time to reach equilibrium. Through the analysis of adsorption kinetics, it could be determined that the adsorption behavior of CV by the SFP/SA hydrogel microsphere mainly conformed to physical adsorption or chemical adsorption.

To study the diffusion mechanism in the adsorption process, the Intraparticle diffusion model, Elovich model and Liquid film diffusion model were used to study the adsorption of CV by SFP/SA, respectively.

At 25 °C, the freeze-dried SFP/SA hydrogel microsphere was placed in water for swelling for 24 h. The CV solution (100 mg/L, pH 7.0) was prepared. Each SFP/SA sample (25 mg) was taken to adsorb 25 mL of CV solution. Then, the absorbance of the CV solution was detected at 585 nm every 20 min.

The equations used are as follows:

The Pseudo-first order model is given as [14]:(4)$$\frac{{\mathrm{dq}}_{t}}{d_{t}}{=k}_{1}{(q}_{e}\;{-\;q}_{t})$$
where q_e_ (mg/g) and q_t_ (mg/g) are the adsorption capacities at equilibrium and at t time, respectively. k_1_ (min^−1^) is the rate constant in the pseudo-first-order kinetic model. The value of k_1_ can be found from the linear plot of log (q_e_ − q_t_) versus time.

The Pseudo-second order model is given as [15]:(5)$$\frac{{\mathrm{dq}}_{t}}{d_{t}}{=k}_{2}{{(q}_{e}\;{-\;q}_{t})}^{2}$$
where q_e_ (mg/g) and q_t_ (mg/g) are the adsorption capacity at equilibrium and at t time, respectively. k_2_ (g mg^−1^ min^−1^) is the pseudo-second-order rate constant. By plotting t/q_t_ as a function of time t, the values of q_e_ and k_2_ were calculated.

The Intraparticle diffusion model is given as [16]:(6)$$q_{t}{=k}_{i}t^{0.5}+C$$
where k_i_ (mg g^−1^ min^−1/2^) is the Intraparticle diffusion rate. C gives information about the thickness of the boundary layer.

The Elovich model is given as [17]:(7)$$q_{t}=\frac{1}{\beta }\ln(\alpha \beta t)$$
where q_t_ (mg/g) is the adsorption capacities at equilibrium and at t time. α (mg g^−1^ min^−1^) implies the initial absorbance. β (g/mg) is the CV desorption rate.

The Liquid film diffusion model is given as [18]:(8)$$-\ln\left(1\;-\;F\right){=k}_{f\;}t\;F=\frac{q_{t}}{q_{m}}$$
where k_f_ (min^−1^) is the Liquid film diffusion coefficient. t (min) is adsorption time. F shows the saturation of adsorption of CV by SFP/SA in Equation (9). q_t_ (mg/g) and q_m_ (mg/g) are the adsorption capacities at time t and equilibrium, respectively.

#### 2.5.3. Adsorption Isotherm Model of CV by SFP/SA

The relationship between adsorption capacity and equilibrium concentration was established by the isothermal adsorption experiment. According to the experimental results of adsorption kinetics, the SFP/SA-60 hydrogel microsphere was selected for the study.

At 25 °C, the freeze-dried SFP/SA-60 hydrogel microsphere was placed in water for swelling for 24 h. The concentrations of the CV solutions (pH 7.0) were 50, 100, 150, 200, and 250 mg/L, respectively. The SFP/SA hydrogel microsphere (25 mg) was taken to adsorb 25 mL of CV solution, and the equilibrium concentration of the CV solution was detected.

The interaction between the SFP/SA hydrogel microsphere and CV was studied by four isothermal models: the Langmuir isotherm model, Freundlich isotherm model, Temkin isotherm model and Dubinin–Radushkevich (D-R) model.

The Langmuir isotherm model is given as [19]:(9)$$\frac{1}{q_{e}}=\frac{1}{q_{m}}+\frac{1}{K_{L}q_{m}}\;\times \;\frac{1}{C_{e}}$$
where q_m_ (mg/g) is the maximum adsorption capacity. C_e_ (mg/L) is the concentration at equilibrium. K_L_ (L/mg) is the Langmuir constant, which represents the energy of adsorption. The plot of 1/q_e_ as a function of 1/C_e_ allows the determination of q_m_ and K_L_.

The separation factor R_L_ is given as:(10)$$R_{L}=\frac{1}{{1+K}_{L}C_{0}}$$

The constant R_L_ indicates the favorability of the adsorption process. If R_L_ = 0, the adsorption is irreversible. If 0 < R_L_ < 1 the adsorption is favorable. The adsorption profile is linear when R_L_ = 1 (K_L_ = 0). When R_L_ > 1 (K_L_ < 0), the adsorption is unfavorable.

The Freundlich isotherm model is given as [20]:(11)$$\log\left(q_{e}\right)\;{=\mathrm{logK}}_{F}+\left(\frac{1}{n}\right){\mathrm{logC}}_{e}$$
where K_F_ (mg/g) and 1/n are the Freundlich constants that indicate the adsorption capacity and the adsorption intensity, respectively.

The Temkin isotherm model is given as [21,22]:(12)$$q_{e}{=B}_{t}{\mathrm{lnK}}_{T}{+B}_{t}{\mathrm{lnC}}_{e}$$

The value of B_t_ (J/mol) (Temkin constant heat of adsorption) and K_T_ (L/mg) (equilibrium binding constant) were computed from the plot of q_e_ and ln C_e_.

The Dubinin–Radushkevich (D-R) isotherm model is given as [23]:(13)$${\mathrm{lnq}}_{e}{=\mathrm{lnq}}_{m}\;{-\;\beta \varepsilon }^{2}\;\varepsilon =\mathrm{RTln}(1+\frac{1}{C_{e}})$$
where q_m_ is theoretical saturation capacity (mg/g). β is an activity coefficient related to mean free energy per mole of adsorbate (mol^2^/J^2^). The value of q_m_ and β are obtained from a linear plot of ln q_e_ vs. ε^2^. The equation for the calculation of “ε” (Polanyi potential) is given in Equation (13). And R is Universal gas constant (8.314 J mol^−1^ K^−1^), T is the temperature in Kelvin.

The E (KJ/mol) value is calculated using the equation:(14)$$E=\frac{1}{\sqrt{2\beta }}$$

If E < 8 KJ/mol, the adsorption is physisorption. If E > 8 KJ/mol, the adsorption is chemisorption.

#### 2.5.4. Desorption and Recycling of SFP/SA

The adsorbed saturated SFP/SA hydrogel microsphere was filtered to remove the surface moisture and then added to 25 mL of 95% ethanol solution. The ethanol solution was replaced every 0.5 h for 2.5 h. After the treatment, the surface ethanol was wiped off and put in the oven at 60 °C for 2 h until completely dry. The above SFP/SA was used to adsorb 100 mg/L of CV solution again to study the reabsorption performance after desorption.

The repeated adsorption rate R_n_ of the hydrogel microsphere was calculated by the following equation:(15)$$R_{n}=\frac{Q_{n+1}}{Q_{n}}$$
where R_n_ is the adsorption rate corresponding to the nth adsorption, Q_n_ is the saturated adsorption capacity corresponding to the nth adsorption (mg/g), and Q_n+1_ is the saturated adsorption capacity corresponding to the nth + 1st adsorption (mg/g).

## 3. Results and Discussion

### 3.1. Analysis of Scanning Electron Microscope

SEM analysis was carried out to see the morphology changes of the internal section of the hydrogel microsphere with the increase of SFP content. The results are shown in Figure 1. The internal structure of the hydrogel microsphere prepared by SA alone (SFP/SA-0) (Figure 1A) showed a tightly layered structure. The formation of the hydrogel microsphere was due to the cross-linking reaction of SA by dropping it into calcium chloride. With the increase in the content of SFP added to the hydrogel microsphere, the layered structure changed from a close structure to a loose structure. The hydrogel micropheres’ internal layer spacing increased notably. At the same time, the internal structure of the hydrogel microsphere transited from a layered structure to a reticulated structure. This transformation gradually made the structure of SFP/SA loose. Therefore, the hydrogel microsphere had a larger space to absorb CV. This transformation increased the contact opportunities between the hydrogel microsphere and CV molecules, which was conducive to the absorption of CV molecules by the hydrogel microsphere. The CV removal ability of hydrogel microsphere in water was ordered as follows: SFP/SA-60 > SFP/SA-40 > SFP/SA-20 > SFP/SA-0.

The morphology changes of the surface before and after CV adsorption were studied by SEM analysis. As shown in Figure 2, the hydrogel microsphere’s surface before CV adsorption had certain wrinkles, which were caused by dehydration during freeze-drying. With the increase of SFP content in the hydrogel microsphere, the wrinkles on its surface gradually decreased. The surface of the hydrogel microsphere became smooth as roughness decreased. This indicated that the addition of SFP could improve the surface smoothness of the hydrogel microsphere by resisting the wrinkles during freeze-drying and providing more sites for CV adsorption. Comparing the appearance of the hydrogel microsphere before (Figure 2A–D) and after (Figure 2E–H) the adsorption of CV, it could be seen that the appearance of the hydrogel microsphere after adsorption became rougher than before. After the hydrogel microsphere adsorbed the CV solution, a large number of CV molecules occupied the active center of the hydrogel microsphere surface. The surface became rough with the increase of surface-adsorbed substances [24].

### 3.2. Analysis of Fourier Transform Infrared Spectroscopy

As shown in Figure 3A, SFP/SA-20, SFP/SA-40, SFP/SA-60 and SFP/SA-80 had similar stretching vibration peaks. Some peaks were observed in structures of SFP/SA-20, SFP/SA-40, SFP/SA-60 and SFP/SA-80 in the range of 2922–2928 cm^−1^, 1623–1632 cm^−1^, 1168–1117 cm^−1^ and 1041–1044 cm^−1^. These vibrations are linked to C-H, C-C, C=O and C-O, respectively [25]. There were antisymmetric stretching vibration peaks and symmetric stretching vibration peaks of -COOH at 1673 cm^−1^ and 1413 cm^−1^, respectively [26]. The SFP curve showed the characteristic peaks at 3460 cm^−1^, 2929 cm^−1^, 1035 cm^−1^ and 818 cm^−1^, which were attributed to -OH stretching, C-H stretching, C-O stretching and glycosidic bond, respectively. Moreover, the antisymmetric stretching of -COOH shifted to 1620 cm^−1^ and the symmetric stretching of -COOH shifted to 1420 cm^−1^. Compared with SFP/SA-0, the SFP/SA-20, SFP/SA-40, SFP/SA-60 and SFP/SA-80 showed unique polysaccharide characteristics. The peaks of SFP/SA-20, SFP/SA-40, SFP/SA-60 and SFP/SA-80 were shifted compared to that of SFP, which may be due to the interaction between SFP and the hydrogel microsphere, indicating that SFP was successfully loaded onto the hydrogel microsphere.

The stretching vibration peak at 900 cm^−1^ was attributed to the stretching of the glycosidic bond [27]. Figure 3B showed that SFP/SA-80, which adsorbed CV, had a stretching vibration peak stretched by the glycosidic bond at 818 cm^−1^, while SFP/SA-80, which did not adsorb CV, did not show a stretching vibration peak. CV and SFP/SA-80 which absorbed CV had C=O stretching vibration peaks around 1170 cm^−1^ (CV at 1175 cm^−1^, and SFP/SA-80, which absorbed CV at 1163 cm^−1^), while SFP/SA-80 did not absorb CV, had no stretching vibration peak around 1170 cm^−^^1^.

The SFP/SA-80 that adsorbed CV did not produce new bonds compared with the SFP/SA-80 that did not adsorb CV, and only the shift between peaks was observed. Therefore, the hydrogel microspheres’ mechanism of CV adsorption was physical adsorption.

### 3.3. Analysis of Texture Profile

As shown in Figure 4, the parameters such as hardness, springiness and resilience of gel beads prepared with different SFP concentrations were determined. Hardness was the maximum force required to compress the sample [28]. Figure 4A shows that the hardness of the hydrogel microsphere increased significantly with the increase of SA concentration, indicating that increasing SA concentration could enhance the hardness characteristics of gel beads. Springiness was a measure of how much the structure of the hydrogel was destroyed by initial compression [29]. High springiness appeared when the hydrogel structure was broken into a few grand pieces during the compression process, whereas low springiness resulted from the hydrogel being broken into many little pieces. Cohesiveness reflects the ability of internal molecules or structural elements to combine and maintain their integrity. Figure 4B and C show that the concentration of SFP has the same effects on the hydrogel microsphere’s springiness and cohesiveness. With the increase of SFP concentration, the springiness and cohesiveness of the hydrogel microsphere increased, indicating that SFP was beneficial to the resistance of the hydrogel microsphere to damage and to its ability to maintain its structural stability. As shown in Figure 4D, with the increase of SFP concentration, the resilience of hydrogel microsphere presented an inverted “U” type relationship. It is analyzed that the addition of SFP made the hydrogel microsphere loose and porous, which increased the resilience of the hydrogel microsphere. However, during the test, the extrusion of the test probe destroyed the porous structure inside the hydrogel microsphere. It caused irreversible physical changes to the hydrogel microsphere, and reduced the resilience. In this process, when the hydrogel microsphere had the highest resilience, the SFP content was about 40%.

### 3.4. Analysis of Differential Scanning Calorimetric

The difference in structure and functional groups of SFP/SA might affect the thermal behavior and transition temperature [26]. For SFP/SA-0, between 25–110 °C, the DSC curve showed a flat endothermic peak, indicating the loss of bound water in the hydrogel microsphere (Figure 5). This was consistent with the conclusion of Mohamadnia et al. in the study of ionic crosslinked carrageenan-alginate hydrogel [30]. Between 125–150 °C, an obvious endothermic peak appeared on the DSC curve, and peaked at 135.7 °C. This was a process by which SA was decomposed into stable intermediate products. Corresponding to the fracture of the SA skeleton, adjacent hydroxyl groups were removed in the form of water molecules. At 155.9 °C, a small endothermic peak appeared, indicating that the intermediate product was further decomposed and decarboxylated to release CO_2_. Mohamadnia et al. suggested that the peak may be attributed to the interaction between cation-alginate and two different uronic acid units of alginate [30]. For the hydrogel microsphere loaded with SFP, the first obvious endothermic peak on the DSC curve was at 136 °C. The second endothermic peak gradually decreased and finally stabilized at 155 °C with the increase of SFP content. The results suggested that SFP achieved effective polymerization in the hydrogel microsphere, and the structure of the hydrogel microsphere was changed. No strong exothermic transformation in the DSC curve of SFP/SA was observed. The contour of Figure 5 tended to show a stronger endothermic peak, indicating that, with the increase of SFP content, the thermal stability of the SFP/SA hydrogel microsphere was improved.

### 3.5. CV Adsorption Behavior of Hydrogel Microsphere

The effects of contact time on the adsorption of CV are shown in Figure 6. In the first 200 min, the contact time had a significant effect on the adsorption, but the process slowed down at 300 min and stabilized after 500 min. At this time, most SFP/SA sites were occupied by CV. It could be seen that the adsorption process curves of SFP/SA-60 and SFP/SA-80 after swelling were roughly the same, and the final saturated adsorption amount was stable at about 80 mg/g. The adsorption capacity of SFP/SA-0 was significantly different from other SFP/SAs. This indicates that the addition of SFP enhanced the CV adsorption capacity of the hydrogel microsphere.

The SFP/SA samples before and after swelling were used to adsorb CV. It can be seen in Figure 7 that the adsorption capacities of SFP/SA were different under the conditions of swelling and unsaturation. The times to reach equilibrious adsorption for the hydrogel microsphere before and after swelling were 2 h and 12 h, respectively. However, the equilibrious adsorption capacity of SFP/SA after swelling was increased by 20.6–327.4% compared with that before swelling. The enhancement of adsorption capacity caused by swelling was mainly due to the early entry of water into the hydrogel microsphere, which made the hydrogel microsphere’s internal layered structure expand more completely, and increased the adsorption capacity of CV.

### 3.6. Adsorption Kinetics

As shown in Table 2, the Pseudo-first order model and Pseudo-second order model R^2^ of SFP/SA were all above 0.99. The R^2^ value of pseudo-first order model was higher than that of the Pseudo-second order model, and the theoretical equilibrious adsorption capacity simulated by the Pseudo-first order model was closer to the actual equilibrious adsorption capacity. It indicated that the adsorption of CV on the SFP/SA hydrogel microsphere was more in line with the pseudo-first-order kinetic adsorption. The adsorption mechanism of SFP/SA for CV mainly was physical adsorption, supplemented by chemical adsorption.

To further study the adsorption mechanism of SFP/SA, the Intraparticle diffusion model, Elovich model and Liquid film diffusion model were selected for analysis. For the Intraparticle diffusion model C = 0, the Intraparticle diffusion was a rate-limiting step. When C > 0, both external mass transfer and Intraparticle diffusion were rate-limiting steps. When C < 0, it could be explained that the external film diffusion resistance was caused by the adsorption time lag [16]. The results showed that with the increase of SFP content, the C of SFP/SA changed from a negative to a positive value. However, there was no case of C = 0 (the curve passes through the origin), indicating that intraparticle diffusion was not the only influencing factor. By comparing the R^2^ corresponding to the Elovich model and Liquid film diffusion model with the pseudo-first order model and pseudo-second order model, the adsorption mechanism of SFP/SA for CV was more in line with the first-order kinetic model.

### 3.7. Isothermal Adsorption

The Langmuir isotherm model, Freundlich isotherm model, Temkin isotherm model and Dubinin–Radushkevich isotherm model were used for the fitting analysis of adsorption isotherms. The isothermal adsorption of SFP/SA-60 hydrogel microsphere for the CV solution at 25 °C was studied. The equilibrious data were fitted with the isothermal adsorption model to find the most suitable adsorption model, indicating that the adsorption of CV by the hydrogel microsphere conformed to the Langmuir model. In the Langmuir model, the adsorption of CV occurred in the specific homogeneous position of the adsorbent hydrogel microsphere, and the monolayer adsorption on the hydrogel microsphere was effective. As shown in Table 3, the R_L_ value was between 0 and 1, indicating that the adsorption of CV on the hydrogel microsphere was beneficial [31]. In addition, the n values in the Freundlich model were ranged from 1 to 10, which indicated that the adsorption process was favorable [32]. According to the Dubinin–Radushkevich isothermal model, the E value was lower than 8 kJ/mol. Indicating that the adsorption of CV solution on hydrogel microsphere belonged to the physical mechanism (such as electrostatic force) [33], which was consistent with the results of adsorption kinetics analysis. Temkin model was also applicable to CV adsorption on hydrogel microsphere for R^2^ > 0.99. It could be concluded that the surface of the hydrogel microsphere was non-uniform and the active center had a uniform binding energy [34].

### 3.8. Desorption and Recycling of Hydrogel Microsphere

The adsorption and desorption properties of the SFP/SA-80 hydrogel microsphere were studied to analyze its recycling capacity. Figure 8 showed that the adsorption capacity of the SFP/SA-80 hydrogel microsphere after 95% ethanol desorption was greatly improved. The second equilibrium adsorption capacity of the hydrogel microsphere was significantly enhanced after ethanol treatment, and the adsorption rate reached 129.14%. It was shown that the internal structure of the hydrogel microsphere became loose after ethanol treatment, which facilitated the exposure of sites for binding CV. The above phenomenon enhanced the speed and adsorption capacity of the hydrogel microsphere for CV adsorption. After 10 adsorption and desorption cycles, the repeated adsorption rate of the hydrogel microsphere was above 97%, and the removal rate of CV was reduced by about 2%. Previous studies had shown that Melanized-C-CNF foam lost about 10% removal efficiency after six desorption cycles [16]. Hydrogel of acrylamide/sodiumalginate/2-acrylamido-2-methylpropane sulphonic acid lost about 5% removal efficiency after six adsorption-desorption cycles [24]. Fe_3_O_4_/CD/AC/SA polymer nanocomposites lost about 5% removal efficiency after six adsorption/desorption cycles [35]. EDTA/graphene oxide functionalized corncob lost about 20% removal efficiency after nine adsorption/desorption cycles [36]. SFP/SA had more advantages both in repetition times and repetition effects.

## 4. Conclusions

In this work, an SFP/SA microsphere was prepared by crosslinking the SFP extracted from *S. fusiforme* with SA via the action of CaCl_2_, which was used to adsorb CV solution. SEM, FT-IR, TPA, DSC and other technologies were used to study the structure of the hydrogel microsphere and the physical and chemical properties after treating with different SFP contents. With an increase in SFP content, the hardness of the hydrogel microsphere decreased, and springiness and cohesiveness increased. At the same time, the thermal stability of SFP/SA hydrogel microsphere was improved. Kinetic studies revealed that the adsorption process was a physical adsorption process supplemented by chemical adsorption, and intraparticle diffusion was not the only influencing factor. Isothermal studies revealed that the adsorption of the SFP/SA hydrogel microsphere was mainly monolayer adsorption. In 10 consecutive cycles, the CV molecules were easily desorbed from the loaded hydrogel microsphere polymer with ethanol, and the repeated adsorption rate remained above 97%. Therefore, the results of this research are expected to provide new insights for the dye adsorption field. The SFP/SA hydrogel microsphere could be used as a good adsorbent for the treatment of CV in sewage.

## Figures and Tables

**Figure 1 molecules-27-04686-f001:**
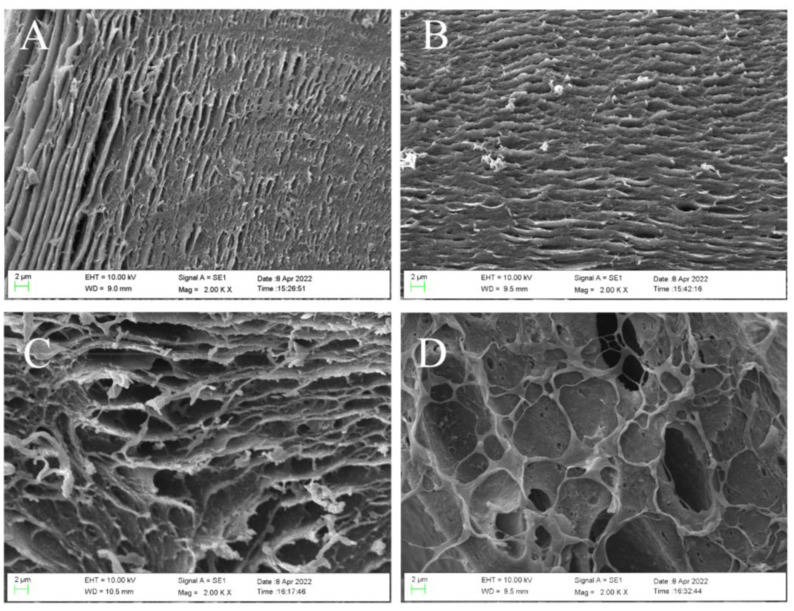
SEM micrograph of internal morphology of SFP/SA-0 (**A**), SFP/SA-20 (**B**), SFP/SA-40 (**C**), and SFP/SA-60 (**D**) at 2000×.

**Figure 2 molecules-27-04686-f002:**
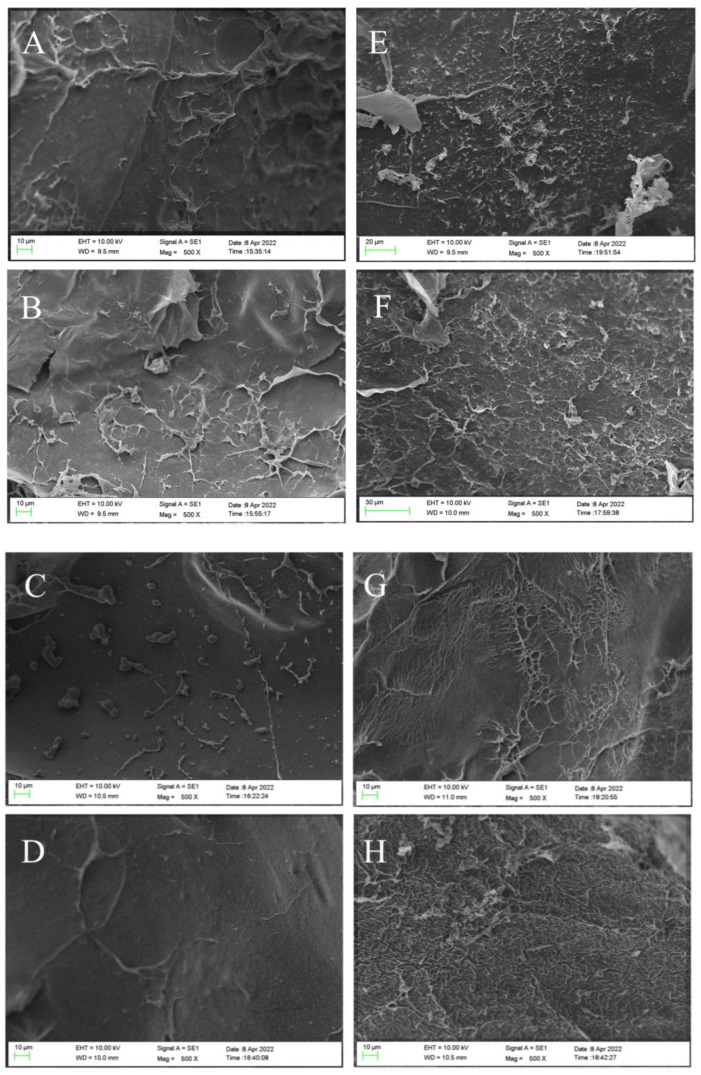
SEM micrograph of surface morphology before absorption of CV SFP/SA-0 (**A**), SFP/SA-60 (**B**), SFP/SA-40 (**C**), and SFP/SA-60 (**D**) and after absorption of CV SFP/SA-0 (**E**), SFP/SA-20 (**F**), SFP/SA-40 (**G**), and SFP/SA-60 (**H**) at 500×.

**Figure 3 molecules-27-04686-f003:**
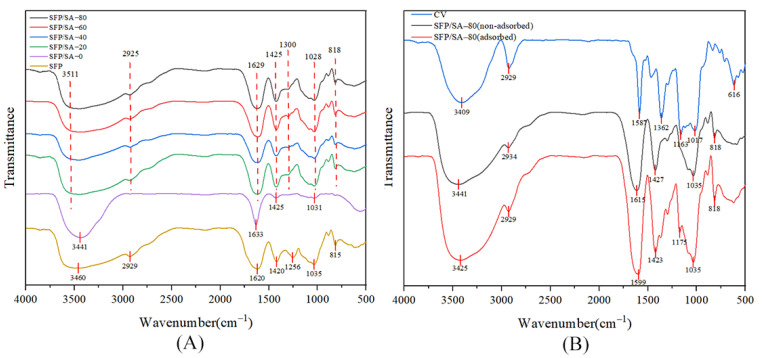
(**A**) The FT-IR spectra of SFP (orange line), SFP/SA-0 (pule line), SFP/SA-20 (green line), SFP/SA-40 (blue line), SFP/SA-60 (red line) and SFP/SA-80 (black line) from 4000–500 cm^−1^. (**B**) The FT-IR spectra of CV (blue line), SFP/SA-80 non-adsorbed (black line) and SFP/SA-80 adsorbed (red line) from 4000–500 cm^−1^.

**Figure 4 molecules-27-04686-f004:**
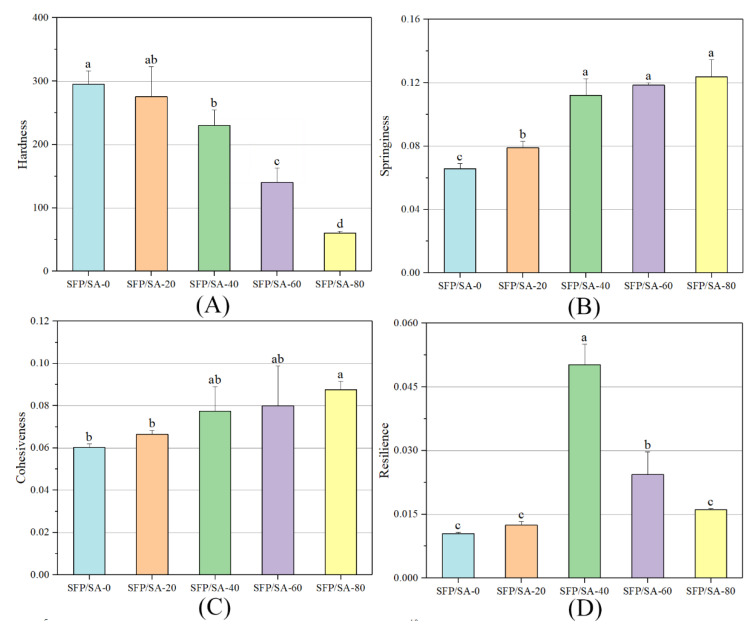
The TPA diagram about SFP/SA hydrogel microspheres of hardness (**A**), springness (**B**), cohesiveness (**C**) and resilience (**D**). Differences between means with the same exponent letter are not significant at a *p*-level value < 0.05.

**Figure 5 molecules-27-04686-f005:**
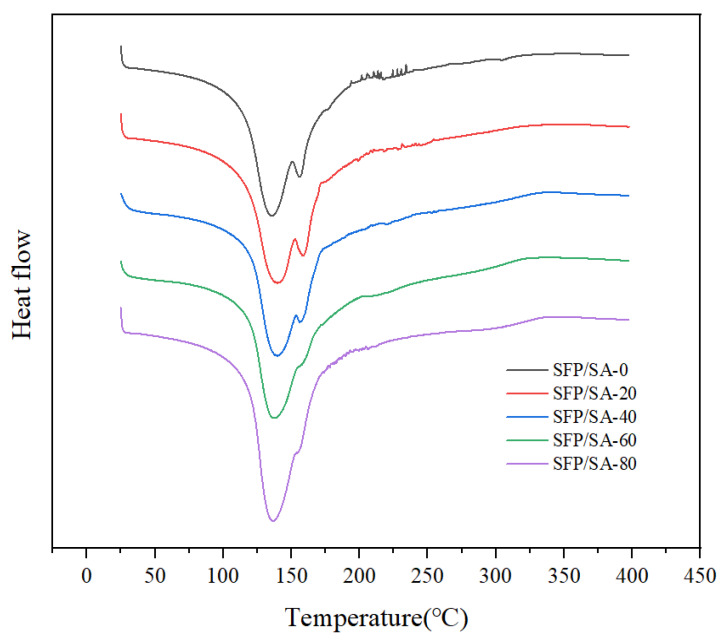
DSC curves of SFP/SA-0 (black line), SFP/SA-20 (red line), SFP/SA-40 (blue line), SFP/SA-60 (green line) and SFP/SA-80 (purple line) from 25–400 °C at a heating rate of 20 °C/min under N_2_ flow.

**Figure 6 molecules-27-04686-f006:**
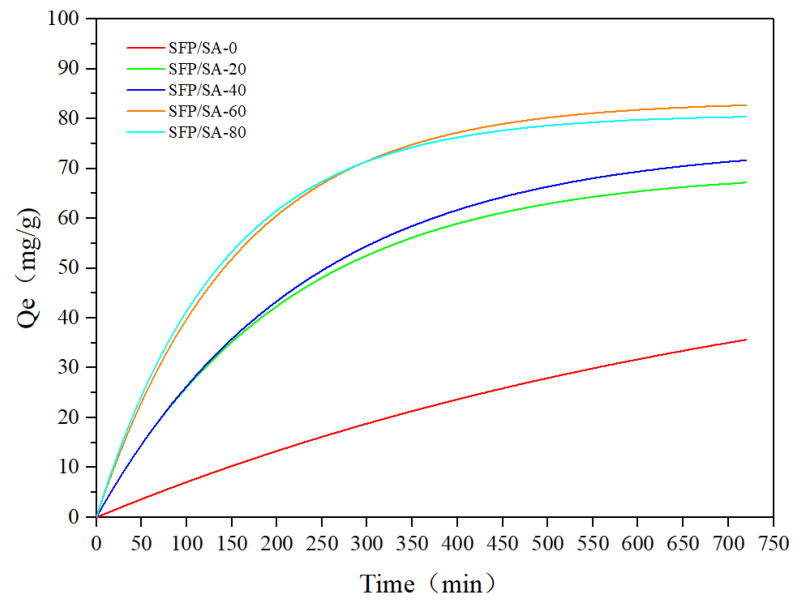
Effect of contact time of SFP/SA on adsorption capacity of CV. (SFP/SA: 25 mg, swelling treatment for 24 h. CV: 25 mL, pH = 7.0, 100 mg/L).

**Figure 7 molecules-27-04686-f007:**
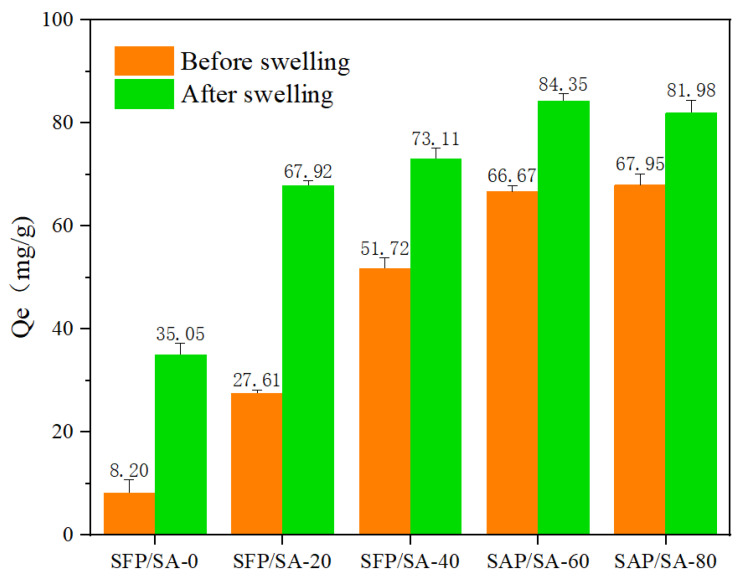
Research of the SFP/SA samples before and after swelling for adsorption capacity.

**Figure 8 molecules-27-04686-f008:**
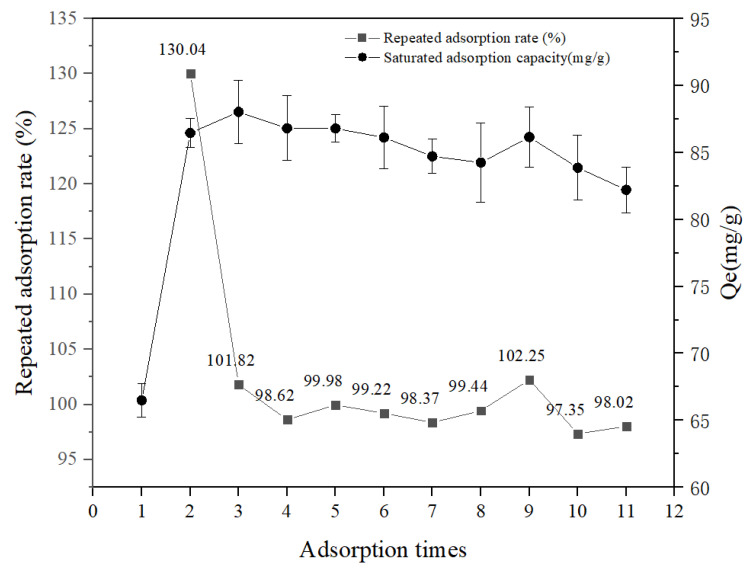
Reusability tests of SFP/SA-80 hydrogel microsphere for CV adsorption.

**Table 1 molecules-27-04686-t001:** Different contents of polysaccharides in SFP/SA.

Sample	SFP (wt%)	SA (wt%)
SFP/SA-0	0	100
SFP/SA-20	20	80
SFP/SA-40	40	60
SFP/SA-60	60	40
SFP/SA-80	80	20

**Table 2 molecules-27-04686-t002:** Adsorption kinetic parameters for CV onto SFP/SA.

Isotherm Model	Parameter	SFP/SA-0	SFP/SA-20	SFP/SA-40	SFP/SA-60	SFP/SA-80
Pseudo-first order	k_1_ (min^−1^)	0.0026	0.0047	0.0043	0.0065	0.0072
	q_e_ (mg g^−1^)	49.25	69.56	75.00	83.50	80.85
	R^2^	0.996	0.997	0.998	0.998	0.997
Pseudo-second order	k_2_ (g mg^−1^ min^−1^)	$3{.62\;\times \;10}^{-5}$	$4{.63\;\times \;10}^{-5}$	$3{.83\;\times \;10}^{-5}$	$6{.62\;\times \;10}^{-5}$	$8{.16\;\times \;10}^{-5}$
	q_e_ (mg g^−1^)	72.86	91.78	100.14	103.16	97.95
	R^2^	0.996	0.992	0.997	0.993	0.996
Intraparticle diffusion	k_i_ (mg g^−1^ min^−1/2^)	1.84	2.78	2.98	3.03	2.80
	C (mg g^−1^)	−6.05	−0.18	−1.40	11.55	15.34
	R^2^	0.991	0.940	0.964	0.897	0.889
Elovich	α (mg g^−1^ min^−1^)	−0.0478	−0.0074	−0.0058	0.0221	0.0340
	β (g mg^−1^)	0.078	0.049	0.046	0.044	0.047
	R^2^	0.944	0.984	0.987	0.982	0.982
Liquid film diffusion	k_f_ (min^−1^)	0.0052	0.0059	0.0055	0.0061	0.0071
	R^2^	0.905	0.971	0.959	0.992	0.939

**Table 3 molecules-27-04686-t003:** Langmuir, Freundlich, Temkin and Dubinin–Radushkevich model isotherm parameters for CV Adsorption by SFP/SA at 25 °C.

Isotherm Model	Parameter	SFP/SA-60
Langmuir	q_m_ (mg/g)	130.14
	K_L_ (L/mg)	0.015
	R_L_	0.21–0.57
	R^2^	0.9967
Freundlich	K_F_ (mg/g)	5.02
	n	1.67
	R^2^	0.9878
Temkin	B_t_ (J/mol)	31.25
	K_T_ (L/mg)	0.13
	R^2^	0.9929
Dubinin–Radushkevich	E (KJ/mol)	0.084
	q_m_ (mg/g)	81.18
	β (mol^2^/J^2^)	7.06 × 10^−5^
	R^2^	0.9029

## Data Availability

Not applicable.

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
