# Peer review of "Sargassum fusiforme Polysaccharide-Based Hydrogel Microspheres Enhance Crystal Violet Dye Adsorption Properties"

_molecules, 2022, doi:10.3390/molecules27154686_

Round 1

Reviewer 1 Report

The article is good and the research is well developed, only that it is necessary to check the English language.
The authors mention Sargassum fusiforme in the title, and in the abstract, but they do not define it in the introduction, they can develop its description in the paper.
In the conclusions, the authors mention microspheres can be used in wastewater adsorption. They can explain what kind of materials can be adsorbed with the developed microspheres and how clean these waters can be.

Author Response

Detailed Responses to the Editor and Reviewers’ comments

Dear Editor,

On behalf of my co-authors, we thank you very much for giving us an opportunity to revise our manuscript. We much appreciate the editors and reviewers for their positive and constructive comments and suggestions on our manuscript entitled "Sargassum fusiforme polysaccharide based hydrogel microspheres enhance crystal violet dye adsorption properties". (ID: molecules-1818211).

We’ve studied reviewer’s comments carefully and have made revision which marked in red in the manuscript. Attached please find the revised version, which we would like to submit for your kind consideration.

Thank you and best regards.

Yours sincerely,

Bingxue Lv

E-mail: lvbingxue@163.com

Reviewer 1: Comments on molecules-1818211 Sargassum fusiforme polysaccharide based hydrogel microspheres enhance crystal violet dye adsorption properties

Comment 1:

The article is good and the research is well developed, only that it is necessary to check the English language.

Response 1:

Thanks for your comments. We have corrected the English language errors in the article according to your suggestion in the attachment.

Comment 2:

The authors mention Sargassum fusiforme in the title, and in the abstract, but they do not define it in the introduction, they can develop its description in the paper.

Response 2:

Thanks for your comments. We have cited a reference for a further definition of Sargassum fusiforme. Please check on Line 51-52 (page 2) in the revised manuscript.

Reference: Zhang, R.; Zhang, X. X.; Tang, Y. X.; Mao, J. L., Composition, isolation, purification and biological activities of Sargassum fusiforme polysaccharides: A review. Carbohydrate Polymers 2020, 228.

Comment 3:

In the conclusions, the authors mention microspheres can be used in wastewater adsorption. They can explain what kind of materials can be adsorbed with the developed microspheres and how clean these waters can be.

Response 3:

Thanks for your comments. The microspheres we developed are mainly used for the adsorption of dye pollutants in sewage, which can effectively reduce the concentration of dye pollutants in sewage. We have refined our conclusions. Please check on Line 450-452 (page 15) in the revised manuscript.

Thank you for suggesting corrections to our article for grammatical errors while reviewing the article.

Reviewer 2 Report

This manuscript reports SFP(S. fusiforme polysaccharide)/SA(Sodium alginate) hydrogel microspheres could enhance the adsorption of CV(crystal violet) in sewage. The author synthesized various SFP/SA combinations by adjusting the amount of SFP to get optimal combinations. The internal structure and textural properties were tested via FR-IR, SEM, DSC, and TPA. The adsorption properties were investigated by the kinetics and isothermal models. Also, the repetition times and effects for regeneration and reuse performance of SFP/SA were superior to those used in other previous studies. However, some parts need to be addressed by the authors before being considered for publication. Therefore, I recommend the manuscript to be published after minor revisions.

Comment 1:

There are several typos, including the comments below, so the author should check the full text to correct any grammar issues.

In page 2, line 47, ‘fastresponse’ should be corrected to ‘fast response’.

In page 5 equation (8) and line 187, the Liquid film diffusion coefficient Kf and kf should be written identically so that there is no misunderstanding.

In page 7, line 241, ‘in the contentof’ needs space between the words.

Comment 2:

As shown in the Figure 1, the author states that as the content of SFP increases, the hydrogel microsphere becomes more porous and has a reticulated structure, which is confusing because the author argues that an increase in the content of SFP decreases the roughness of the surface. Also, in the case of SEM images of SFP/SA samples before CV adsorption in figure 2, there was no correlation with the images shown in figure 1. In the case of SPF/SA-60, a large mesh structure was shown at the magnification of figure 1, but such a structure was not found locally in figure 2D. The author needs to provide a sufficient explanation for these parts.

Comment 3:

Figure 3(B) shows the FT-IR graph of the SFP/SA-80 samples with or without adsorption of CV. For the C=O stretching vibration peaks, 1175 and 1163 cm-1, it is confused that there is no 1163 cm-1 peak for SFP/SA-80 (adsorbed) sample. The author should clarify the sentence “CV and SFP/SA-80 which absorbed CV had C=O stretching vibration peaks at 1175 cm-1 and 1163 cm-1, while SFP/SA-80 did not absorbed CV had no stretching vibration peak.” to prevent the readers misunderstanding.

Comment 4:

As a result of TPA analysis, resilience, chewiness, and gumminess showed an inverted “U” type relationship. However, when the SFP concentration is 40%, the author does not sufficiently explain why it has the best mechanical properties. In addition, the authors argued that the trends in resilience, chewiness, and gumminess were consistent, except when the SFP concentration was 40%, the trends were not at all the same. This trend should be explained as well.

Comment 5:

Figure 5 shows DSC curves with different concentrations of SFP samples. The authors insist that increasing the SFP content improves the thermal stability of SFP/SA hydrogel microspheres. However, to fully understand the thermal behavior of each of SFP and SA, it is necessary to present the DSC curve of SFP alone.

Comment 6:

In the section 3.8. the author explain that the internal structure of hydrogel microsphere became loose after ethanol treatment. If so, does it mean that structural instability increases as the recycle is repeated? Then, it is curious that what is the maximum number of recycles in which structural instability can be ignored.

Author Response

Detailed Responses to the Editor and Reviewers’ comments

Dear Editor,

On behalf of my co-authors, we thank you very much for giving us an opportunity to revise our manuscript. We much appreciate the editors and reviewers for their positive and constructive comments and suggestions on our manuscript entitled "Sargassum fusiforme polysaccharide based hydrogel microspheres enhance crystal violet dye adsorption properties". (ID: molecules-1818211).

We’ve studied reviewer’s comments carefully and have made revision which marked in red in the manuscript. Attached please find the revised version, which we would like to submit for your kind consideration.

Thank you and best regards.

Yours sincerely,

Bingxue Lv

E-mail: lvbingxue@163.com

Reviewer 2: Comments on molecules-1818211 Sargassum fusiforme polysaccharide based hydrogel microspheres enhance crystal violet dye adsorption properties.

This manuscript reports SFP (S. fusiforme polysaccharide) / SA(Sodium alginate) hydrogel microspheres could enhance the adsorption of CV(crystal violet) in sewage. The author synthesized various SFP/SA combinations by adjusting the amount of SFP to get optimal combinations. The internal structure and textural properties were tested via FR-IR, SEM, DSC, and TPA. The adsorption properties were investigated by the kinetics and isothermal models. Also, the repetition times and effects for regeneration and reuse performance of SFP/SA were superior to those used in other previous studies. However, some parts need to be addressed by the authors before being considered for publication. Therefore, I recommend the manuscript to be published after minor revisions.

Comment 1:

There are several typos, including the comments below, so the author should check the full text to correct any grammar issues.

In page 2, line 47, ‘fastresponse’ should be corrected to ‘fast response’.

In page 5 equation (8) and line 187, the Liquid film diffusion coefficient Kf and kf should be written identically so that there is no misunderstanding.

In page 7, line 241, ‘in the contentof’ needs space between the words.

Response 1:

Thank you for suggesting corrections to our article for grammatical errors while reviewing the article. We have revised the format of references. Please check on Line 46 (page 2), 242 (page 7) and page 5 equation (8) in the revised manuscript.

Comment 2:

As shown in the Figure 1, the author states that as the content of SFP increases, the hydrogel microsphere becomes more porous and has a reticulated structure, which is confusing because the author argues that an increase in the content of SFP decreases the roughness of the surface. Also, in the case of SEM images of SFP/SA samples before CV adsorption in figure 2, there was no correlation with the images shown in figure 1. In the case of SPF/SA-60, a large mesh structure was shown at the magnification of figure 1, but such a structure was not found locally in figure 2D. The author needs to provide a sufficient explanation for these parts.

Response 2:

Thanks for your comments. Figure 1 shows the internal morphology of the hydrogel microsphere and Figure 2 shows the surface morphology of the hydrogel microsphere. After freeze-drying the surface of hydrogel microsphere cannot be observed as a laminar or reticular structure under SEM. Hydrogel microsphere is treated with liquid nitrogen to obtain a complete structure before the internal laminar or reticular structure can be observed under SEM. We have revised the captions of the two figures and have also added relevant content to 3.1. Please check on Line 240-241 (page 7), 253 (page 7) and 269 (page 8) in the revised manuscript.

Comment 3:

Figure 3(B) shows the FT-IR graph of the SFP/SA-80 samples with or without adsorption of CV. For the C=O stretching vibration peaks, 1175 and 1163 cm-1, it is confused that there is no 1163 cm-1 peak for SFP/SA-80 (adsorbed) sample. The author should clarify the sentence “CV and SFP/SA-80 which absorbed CV had C=O stretching vibration peaks at 1175 cm-1 and 1163 cm-1, while SFP/SA-80 did not absorbed CV had no stretching vibration peak.” to prevent the readers misunderstanding.

Response 3:

Thanks for your comments. We have revised the article based on your suggestion to prevent the readers misunderstanding. We have indicated that SFP/SA-80 did not absorb CV had no stretching vibration peak around 1170cm-1. Please check on Line 290-293 (page 9) in the revised manuscript.

Comment 4:

As a result of TPA analysis, resilience, chewiness, and gumminess showed an inverted “U” type relationship. However, when the SFP concentration is 40%, the author does not sufficiently explain why it has the best mechanical properties. In addition, the authors argued that the trends in resilience, chewiness, and gumminess were consistent, except when the SFP concentration was 40%, the trends were not at all the same. This trend should be explained as well.

Response 4:

Thanks for your comments. For the inverted “U” shaped trend in the resilience, it is analysed that the addition of SFP made the hydrogel microsphere loose and porous which increased the resilience of the hydrogel microspheres. However, during the test, the extrusion of the test probe destroys the porous structure inside the hydrogel microspheres, causing irreversible physical changes to the hydrogel microspheres and reducing the resilience. In this process, when the hydrogel microsphere had the highest resilience, the SFP content was about 40%.

We have re-analyzed the properties of the parameters in the TPA. As chewiness describes the chewability of a solid sample, and it is classified as a food property. Therefore, hydrogel microsphere does not need to be analyzed for chewiness. Gumminess is used to describe the viscosity properties of semi-solid samples. However, hydrogel microsphere is solids. It is meaningless to analyze the gumminess of the hydrogel microsphere. Therefore, the analysis of chewiness and gumminess of hydrogel microsphere has been eliminated. Thanks for your comments.

We have added the above explanation to the manuscript. Please check on Line 316-323 (page 9) and Fig. 4 (page 10) in the revised manuscript.

Comment 5:

Figure 5 shows DSC curves with different concentrations of SFP samples. The authors insist that increasing the SFP content improves the thermal stability of SFP/SA hydrogel microspheres. However, to fully understand the thermal behavior of each of SFP and SA, it is necessary to present the DSC curve of SFP alone.

Response 5:

Thanks for your comments. The formation of the hydrogel microsphere is due to the cross-linking reaction of SA by dropping into calcium chloride. With the increase of SFP contents, the SA contents used for cross-linking gradually decreased and the intensity of cross-linking reaction occurred decreased. When the solution is prepared exclusively from SFP, the solution does not form hydrogel microsphere when in contact with calcium chloride. (We have found in many experiments that it is impossible to obtain hydrogel microsphere only prepared from SFP). Considering that we used DSC to study the changes in physical properties of hydrogel microsphere with temperature. Therefore, the SFP/SA-0, SFP/SA-20, SFP/SA-40, SFP/SA-60, and SFP/SA-80 which can be formed into hydrogel microspheres were selected in our study. Among them SFP/SA-0 is used as blank control.

Comment 6:

In the section 3.8. the author explain that the internal structure of hydrogel microsphere became loose after ethanol treatment. If so, does it mean that structural instability increases as the recycle is repeated? Then, it is curious that what is the maximum number of recycles in which structural instability can be ignored.

Response 6:

Thanks for your comments. It could be seen from Fig. 8 that after the first treatment with ethanol, the repeated absorption rate of the hydrogel microsphere adsorbing CV increased to 130.04% and then hovered around 99%. We use ethanol to treat the hydrogel microsphere for the same time, and the subsequent drying treatment can effectively prevent the residue of ethanol. Therefore, ethanol has little effect on the stability of the structure of the gel spheres. We mainly focused on whether SFP/SA hydrogel microsphere has cyclic regeneration capacity and its ability to adsorb CV decreased over ten cycles (or less) compared to the hydrogels studied by others. The Fig. 8 shows that it maintains a high level of stability over 10 cycles. Since I have graduated and left school, so I am not able to test the maximum number of recycles currently. Thanks for your understanding. We have cited a reference for the number of cycles of CV adsorption by hydrogels. Their study continued through six cycles, and lost about 5% removal efficiency after six adsorption-desorption cycles. Please check on Line 427-429 (page 17) in the revised manuscript.

Reference: Rehman, T. U.; Bibi, S.; Khan, M.; Ali, I.; Shah, L. A.; Khan, A.; Ateeq, M., Fabrication of stable superabsorbent hydrogels for successful removal of crystal violet from waste water. Rsc Adv 2019, 9, (68), 40051-40061.
